# Assessment of Shoulder Range of Motion Using a Wireless Inertial Motion Capture Device—A Validation Study

**DOI:** 10.3390/s19081781

**Published:** 2019-04-13

**Authors:** Michael Rigoni, Stephen Gill, Sina Babazadeh, Osama Elsewaisy, Hugh Gillies, Nhan Nguyen, Pubudu N. Pathirana, Richard Page

**Affiliations:** 1Department of Orthopaedics, University Hospital Geelong, Barwon Health, Geelong, Victoria 3220, Australia; mrigoni21@gmail.com (M.R.); stephen.gill2@deakin.edu.au (S.G.); sbabazadeh@gmail.com (S.B.); oelsewaisy@gmail.com (O.E.); hugh.gillies@gmail.com (H.G.); 2Barwon Centre for Orthopaedic Research and Education (B-CORE), St John of God Hospital Geelong, Victoria 3220, Australia; 3Deakin University, Waurn Ponds & Burwood, Victoria 3216 & 3125, Australia; ndn@deakin.edu.au (N.N.); pubudu.pathirana@deakin.edu.au (P.N.P.)

**Keywords:** joint range of motion, inertial measurement unit, IMU, wearable, goniometer, ROM, wireless, shoulder

## Abstract

(1) Background: Measuring joint range of motion has traditionally occurred with a universal goniometer or expensive laboratory based kinematic analysis systems. Technological advances in wearable inertial measurement units (IMU) enables limb motion to be measured with a small portable electronic device. This paper aims to validate an IMU, the ‘Biokin’, for measuring shoulder range of motion in healthy adults; (2) Methods: Thirty participants completed four shoulder movements (forward flexion, abduction, and internal and external rotation) on each shoulder. Each movement was assessed with a goniometer and the IMU by two testers independently. The extent of agreement between each tester’s goniometer and IMU measurements was assessed with intra-class correlation coefficients (ICC) and Bland-Altman 95% limits of agreement (LOA). Secondary analysis compared agreement between tester’s goniometer or IMU measurements (inter-rater reliability) using ICC’s and LOA; (3) Results: Goniometer and IMU measurements for all movements showed high levels of agreement when taken by the same tester; ICCs > 0.90 and LOAs < ±5 degrees. Inter-rater reliability was lower; ICCs ranged between 0.71 to 0.89 and LOAs were outside a prior defined acceptable LOAs (i.e., > ±5 degrees); (4) Conclusions: The current study provides preliminary evidence of the concurrent validity of the Biokin IMU for assessing shoulder movements, but only when a single tester took measurements. Further testing of the Biokin’s psychometric properties is required before it can be confidently used in routine clinical practice and research settings.

## 1. Introduction

Accurately assessing joint range of motion (ROM) is integral in clinical orthopaedics and research settings. Several methods and instruments are available for measuring ROM, varying from visual estimation through to specialised kinematic assessment laboratories [1]. Each method has benefits and limitations, with the universal goniometer being the most commonly adopted technique due to portability and ease of use [1]. Goniometers, when used correctly, can accurately measure ROM, however measurement quality is influenced by the tester’s manual skills and methods used [2,3]. 

In recent years, the popularity of commercially available motion tracking devices has increased the use of wearable motion capture systems to measure ROM. Inertial measurement units (IMUs) are one type of motion tracking device that have been widely adopted due to their ease of use, relative low cost, and portability [3,4,5]. These devices utilise recent advancements in the miniaturisation of motion capturing sensors to produce a light, non-invasive and wireless instrument that has the potential to assess human movement in a variety of environments [6,7]. 

The shoulder complex is one of the most complicated joint systems in the body, incorporating the glenohumeral, acromioclavicular, scapulothoracic, and sternoclavicular joints [8]. The shoulder’s design allows for tri-planar upper limb movements, which due to the unique structure and coupled movements that form scapulo-humeral rhythm, produce kinematics that cannot be accurately captured when compared to traditional mechanical or robotic joints [9,10]. For these reasons measuring upper limb kinematics is regarded as the most difficult problem in human motion estimation [5]. However, this challenge must be addressed because measuring shoulder movements is important for assessing upper limb movement, which has obvious clinical significance. If clinically suitable IMUs were available to replace expensive, difficult to access kinematic labs, it would help clinicians better understand the impact of their interventions in routine clinical practice.

Wearable IMU devices are limited by inertial sensor drift and linear acceleration interference [11,12]. Inertial sensors measure segment orientation indirectly by integrating acceleration and angular velocity signals; during this process small errors accumulate over time, which is termed inertial sensor drift [13]. Errors also occur during relatively large linear acceleration, which is termed linear acceleration interference [12]. Investigating human kinematic using accelerometers or gyroscopes alone have been limited by the aforementioned challenges [11,14,15]. To overcome these issues, a tri-axial gyroscope, accelerometer and magnetometer have been combined into a single device and accompanied by a fusion algorithm [6,12,14]. In theory, combining data from each sensor can reduce measurement error and provide a more accurate estimation of motion [14,15]. 

Few studies have directly compared IMU and goniometer measurements of shoulder ROM. Yoon et al. compared IMU and goniometer measurements of static shoulder positions and found high levels of agreement for some positions (i.e., 95% LOA < ±5°), but not all [16]. Agreement reduced at higher levels of shoulder elevation, indicating heteroscedasticity. A recent review by Garimella et al. looked at the accuracy of portable and inexpensive motion capture devices for measuring joint angles compared to benchmark systems, typically optical systems [17]. The investigators demonstrated that from 2009–2017, IMUs were the most popular devices investigated, and found mean average error ranged 0.8°–5.0° for upper limb measurements. Measurement accuracy was dependent on the joint under measurement; joints with large range of movement, such as the shoulder, typically had larger measurement error. The IMU’s underlying algorithm, which combines data from individual components and filters noise, played a significant role in measurement accuracy, and varied between devices. Given that almost 50% of the devices reviewed in the study were from the same developer, Gerimella et al. recommended that the accuracy of other devices also be evaluated.

‘Biokin’ is a locally-developed ROM measurement IMU device that combines tri-axial data from a gyroscope, accelerometer and magnetometer to assess motion [18]. It is small, light-weight and wearable, and can be used for measuring various limb movements. We have previously demonstrated that the IMU can accurately measure wrist movements [18], but its ability to accurately measure ROM at other joints needs to be established. 

The primary aim of this study was to investigate whether the Biokin IMU can accurately measure active shoulder movements in a typical clinical environment. Specifically, we compared Biokin shoulder ROM measurements to universal goniometer measurements in healthy adults. A secondary aim was to investigate the inter-rater reliability of Biokin and goniometer measurements.

## 2. Materials and Methods

### 2.1. Participant Recruitment

Participants were hospital staff or students from our institution. Participants were volunteers and were recruited via communal bulletin boards and posters. Participants were excluded if they had active upper limb pathology and/or symptoms such as pain when moving their arm.

### 2.2. Data Collection

Demographic information was collected from all participants including age, sex, handedness, occupation, sport participation, and prior shoulder injuries.

Data was collected in a hospital orthopaedic ward. Four active movements were tested on each participant’s right and left shoulder according to a standardised protocol: abduction; flexion; and internal rotation and external rotation at 90 degrees of shoulder abduction (see Table 1 and Table 2 for starting positions, and Krishnan et al. for diagrammatic representations of each movement) [10]. Prior to each test, a researcher (MR or HG) demonstrated the movement and the participant practiced the movement until the researcher was satisfied it was performed correctly. Participants were instructed to move the arm in each direction as far as they comfortably could.

One researcher (MR or HG) collected goniometer measurements from each participant and during each movement, IMU measurements were also taken. A second researcher (MR or HG), who was blind to the first measurements, repeated the process. One goniometer and one IMU measurement was taken by each researcher for each movement on each arm.

The IMU was calibrated prior to ROM testing according to the device instructions. The IMU was securely attached to the participant’s forearm with a self-adhering strap, 10 cm distal to the lateral epicondyle (see Figure 1). Measurements were sent wirelessly to a mobile phone and subsequently transferred to Biokin specific software to calculate ROM which was performed by a researcher (NN or PP) who was blind to goniometer measurements. 

### 2.3. Data Analysis

Agreement between IMU and goniometric measurements for each movement on each limb was assessed using intraclass correlation coefficients (ICC) and Bland-Altman analysis [19]. Inter-rater reliability was assessed by comparing each researcher’s IMU and goniometer measurements for each movement using ICCs and Bland–Altman analysis.

ICCs (2,1) were calculated using a two-way random effects model. The magnitude of correlation required to ensure adequate reliability is contested, and clinically acceptable correlations have been suggested anywhere from 0.75 to 0.90 [20,21]. We defined ICCs of less than 0.5 as indicative of poor reliability, values between 0.5 and 0.75 as indicative of moderate reliability, values between 0.75 and 0.90 as indicative of good reliability, and values greater than 0.90 as indicative of excellent reliability [22].

Bland–Altman analysis determines the limits of agreement (LOA) between two measurements [23]. Judgement is required to determine the clinical relevance of the results [24]. We considered a clinically significant change in ROM to be at least 10 degrees, which is consistent with suggestions by others [25]. Consistent with other investigators, acceptable agreement between measurements required the LOA to be within five degrees of no difference between measurements [16].

Assuming alpha of 0.05, power of 0.80, two observations per movement, and an ICC 0.8–0.9, a minimum of 46 observations per movement were required [26]. We considered it feasible to assess 30 participants, where each participant would provide two measurements for each movement direction (e.g. one flexion measurement from each shoulder), producing 60 measurements for each movement direction.

### 2.4. Ethical Considerations

Ethical approval was provided by the organisation’s Human Research Ethics Committee (ref: 16/53). All participants provided informed written consent.

## 3. Results

Participant characteristics can be found in Table 3. Twelve participants were nurses, nine were doctors, four were students, and the remaining were other hospital staff. All participants completed all measurements.

### 3.1. IMU versus Goniometer Measurements

ICCs demonstrated excellent reliability (ICC > 0.90) between IMU and goniometer measurements for all shoulder movements (see Table 4). The mean difference between IMU and goniometer measurements was less than one degree for all movements. The difference in IMU and goniometer measurements was within five degrees of the mean difference for approximately 95% of participants.

### 3.2. IMU versus IMU and Goniometer versus Goniometer Measurements

ICCs ranged from 0.71 to 0.89, indicating moderate to good reliability when each tester’s measurements were compared for the IMU and goniometer (see Table 5 and Table 6). Mean differences between each tester ranged between 0.9 and 5.2 degrees, and limits of agreement were wide, consistently greater than 10 degrees either side of the mean difference in each tester’s scores. 

## 4. Discussion

Examining the reliability of the measuring system is necessary to estimate measurement precision [26]. The current study compared Biokin IMU and goniometer measurements for active shoulder ROM and found high levels of reliability and agreement, providing evidence of concurrent validity of the IMU for measuring shoulder ROM. High agreement suggests that a single assessor’s IMU or goniometer measurements could be used interchangeably. However, inter-rater reliability and agreement was considerably lower, suggesting that one assessor’s measurements cannot be exchanged for another’s measurements for either the IMU or goniometer.

In the current study, LOAs were within our predefined limits and reliability estimates were consistently higher when the same tester took measurements than when different testers took measurements. Most studies of shoulder ROM using goniometers have found intra-rater reliability greater than inter-tester reliability (see Norkin and White for a summary) [25]. Potential sources of error for goniometer and IMU measurements include differences in starting positions and participant effort. Kebaetse et al. found shoulder abduction reduced by 23.6 degrees when participants’ trunks were slouched versus erect [27]. Subtle differences in goniometer arm alignment can also lead to errors in repeated measurements [27]. When assessing change over time, it is recommended to use the same assessor where possible so that differences in measurement values reflect real change rather than measurement error [25]. 

Our results are similar to Yoon et al. as one of few studies that compared IMU and goniometer measurements for shoulder ROM [16]. Unlike our study, Yoon et al. compared measurements for static shoulder positions rather than movements through range. In combination, these studies provide initial evidence regarding the utility of IMUs for accurately measuring shoulder ROM.

Assessing ROM is a fundamental component of a musculoskeletal examination [16]. In clinical settings, the goniometer is commonly used to assess ROM, but is dependent on assessor availability and skill to manually assess and record measurements. Wearable motion capture systems, such as the Biokin, can potentially improve the ease of collecting measurements, and increase the amount and type of data collected. We tested an IMU in a typical clinical environment and found the device could be fitted to a participant in just a few seconds. Fitting the IMU needed one anatomical landmark to be located (lateral epicondyle), whereas the goniometer required three (axis and a landmark for each goniometer arm), which might reduce the IMU’s measurement error. However, reliability and agreement estimates were similar for the IMU and goniometer in our study.

The relative simplicity of fitting and wearing the IMU could allow participants to fit and use the device by themselves, thereby increasing the variety of contexts that ROM could be collected when compared to a goniometer. The Biokin could allow clinicians or researchers to remotely monitor participants’ movements during daily activities, work, or sports; other IMUs have been used to collect shoulder movement in the participant’s workplace [28]. IMUs could also provide participants with feedback on their performance and progress during rehabilitation and help clinicians monitor and better understand the impact of their interventions. Assessing agreement between measurements when an IMU is used independently by a participant than when compared to a clinician or researcher is present is a necessary subject of future research.

Our study has limitations, which represent opportunities for further research. First, the patient cohort consisted of healthy participants, and our results might not be replicated in those with shoulder pathology. Second, each tester took only one measurement with each modality, and intra-rater reliability for each modality could not be determined, nor the effect of averaging measurement across multiple attempts. Third, it is uncertain how these results relate to other joints or more complex movement patterns, such reach-to-grasp upper limb movements.

## 5. Conclusions

Wearable motion capture devices are becoming increasingly common as sensor technology advances. Biokin is one type of wearable IMU that combines tri-axial data from a miniaturised gyroscope, accelerometer, and magnetometer to assess motion. In this study we provide evidence for the concurrent validity of IMU measurements of shoulder ROM compared to goniometer measurements when taken by a single tester. A greater understanding of the psychometric properties of IMUs for assessing different movements by a variety of testers and in different contexts is required before they can be confidently used in routine clinical practice.

## Figures and Tables

**Figure 1 sensors-19-01781-f001:**
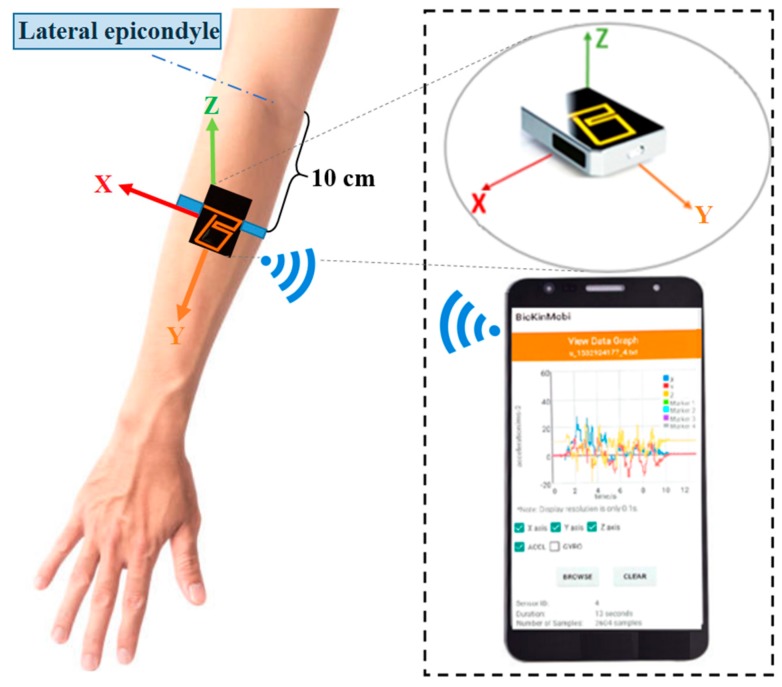
IMU placement.

**Table 1 sensors-19-01781-t001:** IMU ROM testing: movements and starting positions.

Shoulder Movement	Upper Limb Starting Position		
	Shoulder	Elbow	Forearm
**Flexion**	0°	0°	NA
**Abduction**	0°	0°	NA
**External Rotation**	90° abduction	90°	Parallel to floor
**Internal Rotation**	90° abduction	90°	Parallel to floor

**Table 2 sensors-19-01781-t002:** Goniometer ROM testing: movements, starting positions, and landmarks.

Shoulder Movement	Upper Limb Starting Position			Goniometer Landmarks		
	Shoulder	Elbow	Forearm	Axis	Stationary Arm	Moving Arm
**Flexion**	Anatomical position	Anatomical position	Anatomical position	Midpoint lateral aspect acromion	Parallel to midline of trunk	Pointing to lateral humeral epicondyle
**Abduction**	Anatomical position	Anatomical position	Anatomical position	Anterior aspect acromion	Parallel to midline of sternum	Pointing to lateral humeral epicondyle
**External Rotation**	90° abduction	90° flexion	Parallel to floor	Olecranon process ulna	Perpendicular to floor	Pointing to ulna styloid process
**Internal Rotation**	90° abduction	90° flexion	Parallel to floor	Olecranon process ulna	Perpendicular to floor	Pointing to ulna styloid process

Note: All movements occurred with participant standing and the participant’s limb was fully exposed.

**Table 3 sensors-19-01781-t003:** Participant characteristics.

Number	Age (Years) (Average, Range)	Female	Right Hand Dominant	Sport Participation (Average Per Week, Range)	Past Shoulder Injuries (No. Participants)
30	32.8 >(24–62)	18 >(60%)	26 >(86.6%)	2.6 >(0–7)	7 >(23.3%) ^1^

^1^ rotator cuff and labral tears, AC joint and gleno-humeral joint dislocation, frozen shoulder and scapula/humeral open reduction internal fixation post trauma

**Table 4 sensors-19-01781-t004:** Intra-rater reliability indicators for goniometer versus IMU measurements.

	Goniometer Average (SD)	IMU Average (SD)	Difference ^1^ (SD)	ICC (95% CI)	Limits of Agreement ^2^
**Flexion**	155.1 >(14.6)	155.1 >(14.1)	0.0 >(1.6)	0.99 >(0.99–0.99)	−3.2, 3.2
**Abduction**	151.4 >(18.6)	152.2 >(17.8)	−0.8 >(1.9)	0.99 >(0.99–0.99)	−4.5, 2.9
**Internal Rotation**	51.9 >(17.5)	52.8 >(16.8)	−0.9 >(1.7)	0.99 >(0.99–0.99)	−4.2, 2.4
**External Rotation**	89.2 >(17.7)	89.5 >(17.2)	−0.3 >(1.5)	0.99 >(0.99–0.99)	−3.3, 2.7

n = 120 measurements per method; ^1^ Difference = (goniometer – IMU measurement); ^2^ Limit of agreement = (mean difference ±1.96 (SD difference)).

**Table 5 sensors-19-01781-t005:** Inter-rater reliability indicators for goniometer: Tester A versus Tester B.

	Tester A Average (SD)	Tester B Average (SD)	Difference ^1^ (SD)	ICC (95% CI)	Limits of Agreement ^2^
**Flexion**	157.5 >(14.0)	152.7 >(14.8)	4.9 >(7.0)	0.88 >(0.81–0.93)	−8.8, 18.6
**Abduction**	152.9 >(19.2)	149.9 >(18.0)	−3.0 >(8.7)	0.89 >(0.82–0.93)	−20.1, 14.2
**Internal Rotation**	52.4 >(18.0)	51.3 >(17.1)	−0.9 >(13.3)	0.71 >(0.56–0.82)	−27.1, 25.2
**External Rotation**	90.3 >(17.6)	88.2 >(17.9)	−2.2 >(9.9)	0.84 >(0.75–0.90)	−21.7, 17.3

n = 60 measurements per Tester; ^1^ Difference = (Tester A – Tester B); ^2^ Limits of agreement = (mean difference ±1.96 (SD difference)).

**Table 6 sensors-19-01781-t006:** Inter-rater reliability indicators for IMU: Tester A versus Tester B.

	Tester A Average (SD)	Tester B Average (SD)	Difference ^1^ (SD)	ICC (95% CI)	Limits of Agreement ^2^
**Flexion**	152.5 >(14.4)	157.7 >(13.4)	−5.2 >(6.9)	0.88 >(0.80–0.92)	−18.8, 8.3
**Abduction**	150.6 >(17.1)	153.8 >(18.5)	−3.2 >(8.6)	0.88 >(0.81–0.93)	−20.2, 13.7
**Internal Rotation**	52.1 >(16.2)	53.5 >(17.6)	−1.5 >(12.9)	0.71 >(0.56–0.82)	−26.8, 23.8
**External Rotation**	88.4 >(17.3)	90.7 >(17.1)	−2.3 >(9.9)	0.84 >(0.74–0.90)	−21.6, 17.1

n = 60 measurements per tester; ^1^ Difference = (Tester A – Tester B); ^2^ Limit of agreement = (mean difference ±1.96 (SD difference)).

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
