# Peer review of "Assessment of Shoulder Range of Motion Using a Wireless Inertial Motion Capture Device—A Validation Study"

_sensors, 2019, doi:10.3390/s19081781_

Round 1
Reviewer 1 Report
The authors have used a specific IMU to measure the shoulder range of motion. The results have been compared with those of a goniometer. Overall, the paper is interesting and appropriate for the journal, but there are some concerns:
1- Please review the current literature regarding the topic in the introduction of the paper. The intro of the paper provides general knowledge regarding g IMUs which is not appropriate for this paper. For instance, in the discussion, you have mentioned Yoon et al study. You need to introduce the audience with this study (and similar ones even for other joints) in the intro of the paper and then discuss them with the results you obtained in the discussion. You should not talk about new materials in the discussion.
2- The authors have said:
“The shoulder complex is one of the most complicated joint systems in the body, incorporating
the glenohumeral, acromioclavicular, scapulothoracic and sternoclavicular joints …” I think it is important to also talk about the shoulder rhythm, as well. Shoulder rhythm is one of the most important and intricate concepts when dealing with the human shoulder. I think you may use the following works to better target this concept:
https://www.sciencedirect.com/science/article/pii/S0094114X18310814
https://link.springer.com/article/10.1007/s11517-018-1903-3
3- One general question about using wearable motion sensors is about the relation between coordinate frames of IMUs and those recommended by the ISB. Did you use any calibration to adjust the coordinate frames of IMUs with the ISB frames ( I am not talking about the calibration of Biokin that you explained on page 4 )? Please see the following work for more information:
https://www.sciencedirect.com/science/article/pii/S0021929017300374
4- When it comes to the human shoulder usually three coordinates are considered: Plane of elevation, elevation angle and axial rotation of the humerus. I think it makes more sense to report the requested motions based on these coordinates, as well.
5- Overall, at some points, especially in the discussion, the paper reads like a commercial ad for the used IMU device (e.g. see page 9 line 199). If there is anything specific about this IMU that makes it distinguished from other IMUs you need to be specific about them. Otherwise, in my opinion, you should talk about the potential of IMUs in general and not just the specific device that you used.
Author Response
Please see attached word document.

Reviewer 2 Report
Dear Authors, The presented manuscript about comparison of biokin device and goniometer motion measurements is quite interesting. The methods and results are sufficiently described,, I would suggest however consideration of assessing the repeatability of the measurements for sifnificamtly greater number of iterations with single volunteer - now there is no information about the repeatability of both methods, as the measurements were taken only twice. I believe that this missing information will significantly support the eventual conclusions. Sincerely, Reviewer
Author Response
Please see attached word document.

Reviewer 3 Report
In the manuscript, the authors introduced the IMU 'Biokin' for assessing shoulder range of motion and evaluated the concurrent validity of the assessment. The manuscript is well-presented but the content is lack in significance.
- According to the Introduction, the background of this study is for accurately assessing the shoulder ROM. The authors needs to provide the information on expected clinical shoulder ROM accuracy. The results should be validated whether the sensors can achieve the expected accuracy.
- There are a range of wearable IMUs for biomechanical purposes e.g. Xsens MTx. Can the authors explain why 'Biokin' is selected.
- Have the authors explore how the attachment of the sensors affected the accuracy of the sensing results.
Author Response
Please see attached word document.

Round 2
Reviewer 1 Report
The manuscript is acceptable to this reviewer. Best of luck to the authors in their future works.
Reviewer 2 Report
The paper was improved substantially, more than I expected in my comments.
There are still weak points, but the Authors adress them as requiring further studies.
I have no further comments
sincerely,
Reviewer